# GRADE Use in Evidence Syntheses Published in High-Impact-Factor Gynecology and Obstetrics Journals: A Methodological Survey

**DOI:** 10.3390/jcm12020446

**Published:** 2023-01-05

**Authors:** Hui-Juan Yang, De-Yu Zhang, Ying-Ying Hao, He-Li Xu, Yi-Zi Li, Shuang Zhang, Xin-Yu Li, Ting-Ting Gong, Qi-Jun Wu

**Affiliations:** 1Department of Clinical Epidemiology, Shengjing Hospital of China Medical University, Shenyang 110004, China; 2Clinical Research Center, Shengjing Hospital of China Medical University, Shenyang 110004, China; 3Department of Obstetrics and Gynecology, Shengjing Hospital of China Medical University, Shenyang 110004, China; 4Department of Nephrology, Dalian Municipal Central Hospital, Dalian 116027, China; 5Liaoning Key Laboratory of Precision Medical Research on Major Chronic Disease, Shengjing Hospital of China Medical University, Shenyang 110004, China; 6Key Laboratory of Reproductive and Genetic Medicine, China Medical University, National Health Commission, Shenyang 110004, China

**Keywords:** appraisal tool, certainty of evidence, GRADE, gynecology and obstetrics, systematic reviews

## Abstract

**Objective:** To identify and describe the certainty of evidence of gynecology and obstetrics systematic reviews (SRs) using the Grading of Recommendations, Assessment, Development and Evaluation (GRADE) approach. **Method:** Database searches of SRs using GRADE, published between 1 January 2016 to 31 December 2020, in the 10 “gynecology and obstetrics” journals with the highest impact factor, according to the Journal Citation Report 2019. Selected studies included those SRs using the GRADE approach, used to determine the certainty of evidence. **Results:** Out of 952 SRs, ninety-six SRs of randomized control trials (RCTs) and/or nonrandomized studies (NRSs) used GRADE. Sixty-seven SRs (7.04%) rated the certainty of evidence for specific outcomes. In total, we identified 946 certainty of evidence outcome ratings (*n* = 614 RCT ratings), ranging from very-low (42.28%) to low (28.44%), moderate (17.65%), and high (11.63%). High and very low certainty of evidence ratings accounted for 2.16% and 71.60% in the SRs of NRSs, respectively, compared with 16.78% and 26.55% in the SRs of RCTs. In the SRs of RCTs and NRSs, certainty of evidence was mainly downgraded due to imprecision and bias risks. **Conclusions:** More attention needs to be paid to strengthening GRADE acceptance and building knowledge of GRADE methods in gynecology and obstetrics evidence synthesis.

## 1. Introduction

Systematic reviews (SRs) are essential parts of evidence-based medicine and serve as the basis for clinical practice guidelines [1], which are widely used in the field of gynecology and obstetrics [2,3,4,5,6]. Recently, several publications have concentrated on the quality and credibility of systematic reviews as the number of such publications has increased [7,8].

The GRADE system is an emerging method for appraising studies and making recommendations for systematic reviews and guidelines [9,10]. As one of the most important methodological achievements of evidence-based medicine (EBM) in the last 30 years, it has been used by over 100 organizations up to now [11]. Different from other appraisal tools (The Jadad, the Newcastle-Ottawa score etc.), GRADE separates quality of evidence and strength of recommendation, assesses the quality of evidence for each outcome, and allows observational studies to be “upgraded”, if they meet certain criteria [12]. There are five distinct steps in the GRADE method [9,12]. Step 1: A prior ranking. For randomized controlled trials, we assign a high ranking, and for observational studies, a low ranking. Step 2: ‘Downgrade’ or ‘Upgrade’ the initial ranking. There are five downgrading domains (risk of bias (RoB), inconsistency, indirectness, imprecision, and publication bias) and three upgrading domains (large consistent effect, dose response, and plausible confounding, which would reduce a demonstrated effect). Step 3: Assign a final grade. On the basis of upgrading and downgrading domains in step 2, the final evidence quality is rated “high”, “medium”, “low”, or “very low” [13]. All the three steps above are repeated for each critical outcome. Step 4: Consider factors affecting recommendation. In addition to evidence quality, recommendations must also take other factors into account, such as cost-effectiveness, patient preference, and balance of desirable and undesirable effects. Step 5: Combine the above factors to give a final recommendation, strong or weak [14].

In general, GRADE has a strict procedure for evaluating evidence and considers other factors besides evidence, which makes it more suitable for the medical field. Previous studies have explored the use of GRADE in the fields of nutrition, urology, and nephrology [15,16]. Among these studies, the number of SRs using GRADE was limited, and the certainty of most evidence was quite low. Considering the merits of GRADE in the evaluation of evidence, there is a need for enhancing the acceptance and use of GRADE in both fields. However, no study has evaluated the application of the GRADE approach in the SRs of journals of gynecology and obstetrics. Therefore, it seems sensible to explore the current status of the GRADE approach used in SRs published in gynecology and obstetrics journals. Herein, we take the following two steps: (1) identify and describe all relevant SRs using the GRADE methodology, to evaluate the outcome-specific certainty of evidence published between 2016 and 2020, in the 10 gynecology and obstetrics journals, with the highest impact factor according to the JCR 2019, and (2) summarize and present the GRADE specific information, including the number of outcomes rated, the certainty of evidence ratings, the use of summary of findings tables, down- and upgrading factors, while also taking the study design (SRs of RCTs vs. NRSs) into account.

## 2. Materials and Methods

### 2.1. Search Strategy

Systematic reviews (SR) published between 1 January 2016 to 31 December 2020 in the 10 gynecology and obstetrics journals, with the highest impact factor (range: 17.18–4.25), according to the JCR 2019, were identified through searches in the database PubMed. (Appendix A).

### 2.2. Inclusion and Exclusion Criteria

SRs were included if they met the following criteria: (1) SRs published between 1 January 2016 and 31 December 2020 in the top 10 journals according to the JCR 2019 category gynecology and obstetrics, and (2) SRs applying the GRADE approach to rate the certainty of evidence.

SRs were excluded if they met the following criteria: (1) SRs using a modified version of the GRADE approach, based on the adaptions of the tool to self-defined criteria of authors to assess the quality of evidence, and (2) SRs failing to provide detailed GRADE evaluation processes and results.

### 2.3. Selection Process of Sources of Evidence

First, title and abstract screening were performed by two reviewers (Y-ZL, SZ.) to identify articles relevant to GRADE. Second, for all potentially relevant references, full-text publications were obtained and checked for final inclusion, by two reviewers (H-JY, H-LX.) independently. Discrepancies were resolved through discussion with a third author (Q-J W.).

### 2.4. Data Extraction

For included SRs, one reviewer (H-JY.) extracted the data and an independent reviewer (H-LX.) cross-checked all data. The following data were extracted at SR level: year of publication, journal name, number of primary studies included, type of studies (RCTs vs. NRSs (including: i.e., non-randomized intervention studies, case-control studies, cohort studies, and cross-sectional studies vs. combination RCTs/NRSs) included, number of participants, description of intervention(s)/exposure(s), number and types of outcome(s) and comparison(s) rated, category of certainty of evidence ratings (high, moderate, low, or very low), meta-analysis conducted (yes vs. no), summary of findings table reported (yes vs. no), number of down and/or upgrades (count of the respective downgrading/upgrading domain used at the outcome-level), and reasons for down- and upgrading. For the quantitative presentation, all downgrading factors listed in the SRs of both RCTs and NRSs were extracted according to the study design, except in the case of two SRs, where a differentiation between study designs was not possible.

Finally, in this methodological survey, we excluded two types of SRs based on minimum criteria proposed by the GRADE working group: (1) the authors of SRs rated the certainty of evidence of each individual study (“study level”); and (2) the authors of SRs rated the certainty of the SR’s body of evidence, instead of rating the body of evidence for a given outcome (“outcome level”) [15].

## 3. Results

### 3.1. Search Results and Sample

Appendix A shows the flow diagram of the literature search. The database search retrieved 1033 documents. After the removal of duplicate records (*n* = 2), title and abstract screening (*n* = 79), and analysis of the remaining 952 full-text articles, 93 articles remained. Among them, 3 SRs using a modified version of the GRADE [17,18,19] and 12 SRs failing to provide a detailed evaluation process [20,21,22,23,24,25,26,27,28,29,30,31] were excluded from this study. A total of 11 SRs using GRADE, did not rate the certainty of evidence for the specific outcome [32,33,34,35,36,37,38,39,40,41,42], but instead assessed certainty of evidence of individual studies or overall certainty of the body of evidence (Appendix A). Finally, only 67 SRs (7.04%; out of 952 SRs published) were included in this study.

Table 1 shows the distribution of these SRs according to the journal and year. The publication of SRs in the top 10 gynecology and obstetrics journals decreased between 2016 and 2019 and increased in 2020, with the least SRs published in 2019 (*n* = 160), and the most in 2020 (*n* = 238). As compared to 2016 and 2017 (6.12% and 6.25%), the proportion of SRs rating the certainty of evidence with GRADE increased among all SRs published in 2018, 2019, and 2020 (9.2%, 6.25%, and 7.56%). More than 80% of all included SRs were published in four journals (Ultrasound in Obstetrics & Gynecology, Human Reproduction Update, British Journal of Obstetrics and Gynecology, and American journal of Obstetrics and Gynecology).

### 3.2. Characteristics of the Included SRs

In total, 41 SRs analyzed evidence from RCTs only (*n* = 41) [43,44,45,46,47,48,49,50,51,52,53,54,55,56,57,58,59,60,61,62,63,64,65,66,67,68,69,70,71,72,73,74,75,76,77,78,79,80,81,82,83] and 18 NRSs only [84,85,86,87,88,89,90,91,92,93,94,95,96,97,98,99,100,101], and 8 SRs that analyzed both RCTs and NRSs [102,103,104,105,106,107,108,109] were included into this methodological study. The characteristics of these SRs are summarized in Table 2, Table 3 and Table 4. Furthermore, from the 8 SRs including both RCTs and NRSs, 4 SRs [103,105,108,109] rated the certainty of evidence derived from RCTs and NRSs separately (in separate rows within a single Summary of Findings table, or in separate Summary of Findings tables), 2 SR [102,104] pooled RCTs and NRSs rated the combined evidence (in the same rows within a single Summary of Findings table), 2 SRs [106,107] rated only evidence from NRSs.

Sixty-four SRs (85%) conducted at least one meta-analysis [43,44,45,46,47,48,49,50,51,52,53,54,55,56,57,58,59,60,61,62,63,64,65,66,67,68,69,70,71,72,73,74,75,76,77,78,79,80,81,82,83,84,85,86,87,88,89,91,92,93,94,95,96,97,98,99,102,104,106,107,108,109], and sixty SRs (83%) showed their findings in a summary of findings table [43,44,45,46,47,48,49,50,51,52,53,54,55,56,57,58,60,61,62,63,64,65,66,67,68,69,70,71,72,73,74,75,76,77,80,81,82,84,85,87,88,89,90,91,93,94,95,96,100,101,103,104,105,106,107,109]. The median number of primary studies included in the SRs was 15 (IQR: 8–21). The median number of participants included in the SRs was 3962 (IQR: 1632–7238) in the SRs of RCTs, and 5829 (IQR: 2791–333,396) in the SRs of NRSs, and 2583 (IQR: 260–244,263) in the SRs of both RCTs and NRSs (Table 5). In the identified SRs, interventions/exposures can be categorized into drug therapy (*n* = 28), surgical therapy (*n* = 10), assisted reproductive (*n* = 9), drug and surgical therapy (*n* = 2), screening method (*n* = 4), special disease (*n* = 2), reproductive strategy (*n* = 1), lifestyle factors (*n* = 3), clinical care (*n* = 1), virus (*n* = 1), and others (*n* = 6).

### 3.3. Certainty of Evidence Ratings

The median of the total number of outcomes rated in a SR was 5 (IQR: 3–8). (Table 5) Overall, there were 946 individual outcome ratings: 42.28% of very low, 28.44% of low, 17.65% of moderate, and 11.63% of high, in certainty of evidence. Among 614 outcomes in the SRs of RCTs, 26.55% were rated very low, 32.74% low, 23.94% moderate, and 16.78% high, in certainty of evidence. In the SRs of NRSs (outcomes were 324), 71.60% were rated very low, 20.73% low, 5.86% moderate, and 2.16% high, in certainty of evidence. In the SRs of NRSs and RCTs, a total of 8 outcomes were rated: 5 of very low, 2 of low, and 1 of moderate certainty of evidence. In studies assessing interventions with drug therapy, there were 445 individual outcome ratings: 415 in the SRs of RCTs, 26 in the SRs of NRSs, and 4 in the SRs of NRSs and RCTs. The certainty of evidence was rated as very low (33.26%), low (23.37%), moderate (22.70%) and high (20.67%). Details can be seen in Table 6.

**Table 2 jcm-12-00446-t002:** Summary of the study characteristics and the number and type of (un)rated outcomes of systematic reviews of randomized controlled trials (*n* = 41) that rated the outcome-specific certainty of evidence with GRADE.

Author, Year	Intervention/Exposure	Number of Studies	Number of Participants	Number of Outcomes Rated (Number of Unrated Outcomes)	Number of Rated Comparisons	Outcomes Rated
Wei et al., 2016 [44]	Paroxetine	5	1571	10	1	Mean reduction in frequency of VMS at (week 4; week 6; week 12); Mean reduction in daily VMS severity score at (week 4; week 12); Headache; Dizziness; Nausea; Fatigue/Drowsiness/Somnolence/Lethargy; Constipation
Rydén et al., 2016 [45]	Aromatase inhibitors alone or sequentially combined with tamoxifen	8	unclear	8	1	Disease-free survival; Overall Survival; Death without recurrence; Endometrial cancer; Fractures; venous thromboembolism events; Cerebrovascular events; Cardiovascular events
Nastri et al., 2016 [46]	Embryo culture using Low O2	21	unclear	3 Clinical outcomes	2 (AtmO2 during all embryo culture, AtmO2 after Day 2)	Live birth/ongoing pregnancy; Clinical pregnancy; Miscarriage
				5 Laboratory outcomes	3 (AtmO2 during all embryo culture, before Day 3 followed by LowO2 in both groups, after Day 2)	Fertilization; Cleavage; High/Top cleavage; Blastocyst; High/Top blastocyst
Di et al., 2016 [47]	Hysteroscopy	9	2976	3	2 (Hysteroscopy vs. no hysteroscopy, Operative vs. diagnostic hysteroscopy)	Live birth rate; pregnancy rate; miscarriage rate
Barbosa et al., 2016 [48]	Oral dydrogesterone	8	3809	5 (3)	2 (vaginal progesterone capsules, Vaginal progesterone gel)	Live birth; Ongoing pregnancy; Clinical pregnancy; Miscarriage per clinical pregnancy; Dissatisfaction
Martins et al., 2016 [49]	Gonadotrophin-releasing hormone agonist	10	3056	3	1	Live birth/ongoing pregnancy; Clinical pregnancy; Miscarriage per clinical pregnancy
Tsiami et al., 2016 [50]	Salpingectomy, proximal tubal occlusion, aspiration of the hydrosalpingeal fluid	7	unclear	4	1	Ongoing pregnancy (direct evidence); Ongoing pregnancy (network meta-analysis); Clinical pregnancy (direct evidence); Clinical pregnancy (network meta-analysis)
Kollmann et al., 2016 [51]	Menopausal gonadotropin, metformin, mannitol	66	unclear	4	1	live birth/ongoing pregnancy; OHSS; clinical pregnancy; miscarriage;
Berghella et al., 2017 [52]	Cervical cerclage	5	419	19	1	PTB (<35 w; 37 w; 34 w; 32 w; 28 w; 24 w); GA at delivery (w); Latency (days); PPROM; Birth weight (grams); LBW; VLBW; RDS; IVH; Sepsis; NEC; NICU; LOS in NICU (days); Neonatal death
Berghella et al., 2017 [53]	Cervical length screening	3	287	7 (1)	1	PTB (<37 w; 34 w; 32 w; 28 w); LBW; Perinatal death; Maternal hospitalization
Romero et al., 2017 [54]	Vaginal progesterone	6	909 (303 women, 606 infants)	23	1	PTB (<33 w; 37 w; 36 w; 35 w; 34 w; 32 w; 30 w; 28 w); SPTB (<33 w; 34 w); RDS; NEC; IVH; Proven neonatal sepsis; Retinopathy of prematurity; Fetal death; Neonatal death; Perinatal death; Composite neonatal morbidity/mortality; Birth weight < 1500 g; 2500 g); Admission to NICU; Mechanical ventilation
Bechtejew et al., 2017 [55]	clomiphene and/or letrozole	23	1961	6	2	live birth and OHSS; clinical pregnancy; miscarriage per clinical pregnancy; number of oocytes retrieved; number of FSH ampoules (75 IU); and cycle cancellation
Sjöström et al., 2017 [56]	no-doctors	6	6735	2 (1)	1	effectiveness; acceptability
Pinto-Lopes et al., 2017 [57]	antibiotic	16	2695	4	1	Composite postpartum infectious morbidity; Endometritis; Wound infection; Urinary tract infection
Saccone et al., 2017 [58]	Vaginal progesterone	3	680	12 (1)	1	SPTB < 37 w; SPTB (<34 w; 32 w; 28 w); Adverse drug reaction; Admission to NICU; RDS; BPD; IVH; NEC; Sepsis; Perinatal death (SPTB < 37 weeks)
Luque-Ramírez et al., 2018 [59]	Combined oral contraceptives and/or antiandrogens	33	1521	7	1	Efficiency (Hirsutism; Menstrual dysfunction); Cardiometabolic risk factors (BMI; Abnormal glucose tolerance; Lipid profile; Blood pressure; Hypertension)
Senra et al., 2018 [60]	Gonadotrophin-releasing hormone plus chemotherapy	13	1208	2	1	Primary ovarian insufficiency; Spontaneous pregnancy
Kalafat et al., 2018 [61]	Metformin	15	3124	3	4 (insulin, glyburide, placebo, other drugs or placebo)	Hypertensive disorders of pregnancy; Preeclampsia; PIH
Vitagliano et al., 2018 [62]	Endometrial scratch injury	10	1468	5	1	Live birth rate; Clinical pregnancy rate; multiple PR; miscarriage rate; EPR
Vitagliano et al., 2018 [63]	Endometrial scratch injury	8	1871	5	1	Clinical pregnancy rate; Ongoing pregnancy rate; multiple PR; EPR; miscarriage rate
Siristatidis et al., 2018 [64]	Human papilloma virus	14	2348	8	1	Live birth/ongoing pregnancy; Miscarriage rate per clinical pregnancy; Clinical pregnancy; Positive pregnancy test; ectopic pregnancy; Live birth/ongoing pregnancy-male factor; Clinical pregnancy-male factor; Miscarriage rate-male factor
Gadalla et al., 2018 [65]	Clomiphene citrate	33	4349	4	1	Endometrial thickness; Ovulation; Pregnancy; Live birth
Romero et al., 2018 [66]	Vaginal progesterone	5	974	32	1	Preterm birth (<33 w; 37 w; 36 w; 35 w; 34 w; 32 w; 30 w; 28 w); Spontaneous preterm birth (<33 w; 34 w); GA at delivery; RDS; NEC; IVH; Bronchopulmonary dysplasia; Retinopathy of prematurity; Fetal death; Neonatal death; Perinatal death; Composite neonatal morbidity/mortality; Apgar score < 7 at 5 min; Birthweight (<1500 g; <2500 g); Admission to NICU; Mechanical ventilation; Congenital anomaly; Bayley-III cognitive composite score at age 2 year; Moderate/severe neurodevelopmental impairment at age 2 year; Visual or hearing Impairment at age 2 year; gastrointestinal; or respiratory function at age 2 year; Any maternal adverse event
Matthewman et al., 2018 [67]	Physical activity	15	1681	2	1	Pain intensity; pain duration
Fang et al., 2018 [68]	oil-soluble contrast material	6	2562	4	1	Ongoing pregnancy; Live birth; Miscarriage; ectopic pregnancy
Vitagliano et al., 2019 [69]	Gonadotrophin-releasing hormone	15	2345	2	1	Ongoing pregnancy rate/Live birth rate; Clinical pregnancy rate
Jarde et al., 2019 [70]	Vaginal progesterone, oral progesterone,17-OHPC, cerclage, and pessary	40	11,311	3	1	Preterm birth < 34 weeks; Preterm birth < 37 weeks; Neonatal death
Baiju et al., 2019 [71]	Self-assessment of outcome	4	5493	6-Effectivenes	1	Complete Abortion; Need for Surgery; Hemorrhage (Excessive Bleeding); Fever and Infection; Drugs for hemorrhage; Ongoing pregnancy
				6-Safety	1	Complete Abortion; Need for surgery; Excessive bleeding; Fever and infection; Drugs for hemorrhage; Continuing pregnancy
Bosdou et al., 2019 [72]	Frozen-embryo transfer	8	5265	5	1	Live birth; Ongoing pregnancy; Clinical pregnancy; ovarian hyperstimulation syndrome; Miscarriage
Wang et al., 2019 [73]	Letrozole, clomiphene + metformin	20	3962	5	1	Live birth; Clinical pregnancy; Multiple pregnancy; Miscarriage; Ovulation
Sotiriadis et al., 2019 [74]	Elective induction	5	7261	9	1	Cesarean section; NICU admission; Operative delivery; Grade-3/4 perineal laceration; postpartum hemorrhage; Postpartum maternal infection; Maternal hypertension; Neonatal death; Neonatal respiratory support
Li et al., 2020 [75]	Progestogen	10	5056	9	1	Live birth; Live birth-oral progestogen; Live birth vaginal progesterone; Miscarriage; Miscarriage oral progestogen; Miscarriage vaginal progesterone; Preterm birth; low birth weight; Congenital abnormalities
Qin et al., 2020 [76]	Acupuncture therapies	5	341	3	3 (antibiotics; no treatment; sham acupuncture)	Composite cure rate; Recurrent rate; Symptom duration
Danhof et al., 2020 [77]	intrauterine insemination	26	5316	3	1	Live birth/ongoing pregnancy; multiple pregnancy; clinical pregnancy
Bergeron et al., 2020 [43]	Endometrial ablation or resection	13	884	13	1	Hysterectomy rates; PBAC scores (at 3 months; 6 months; 12 months; 24 months); Amenorrhea (at 6 months; 12 months; 24 month); Side effects; Quality of life; Treatment failures; Satisfaction rate; Hemoglobin
Islam et al., 2020 [78]	Prophylactic antibiotics	24	16,178	2	1	Genital tract infection; adverse event
Bordewijk et al., 2020 [79]	individual participant data sharing	45	8697	2	1	Live birth; Clinical pregnancy
Cai et al., 2020 [80]	Prophylactic antibiotics	20	7169	2	1	Chorioamnionitis; Neonatal Death
Di Mascio et al., 2020 [81]	Delayed pushing	12	5445	17	1	SVD; OVD; CD; OD; CD in the second stage; Duration of second stage; Time of active pushing; Chorioamnionitis; Intrapartum fever; Endometritis; PPH; Episiotomy; Severe perineal lacerations; Low umbilical cord pH; Apgar < 7 at 5 min; Neonatal respiratory morbidity; NICU admission
Samy et al., 2020 [82]	Misoprostol; oxytocin; vasopressin; tranexamic acid; epinephrine; or ascorbic acid	26	1627	1	1	Blood loss
Stewart et al., 2020 [83]	Trastuzumab	5	11,376	2	1	Disease-free survival; Overall Survival

Outcome graded: BMI, Body Mass Index; BPD, bronchopulmonary dysplasia; CD, cesarean delivery; EPR, ectopic pregnancy rate; FSH, follicle-stimulating hormone; GA, gestational age; IVH, Intraventricular hemorrhage; LBW, low birth weight; LOS, length of stay; NEC, necrotizing enterocolitis; NICU, admission to the neonatal intensive care unit; OD, operative delivery; OHSS, ovarian hyperstimulation syndrome; OVD, operative vaginal delivery; PBAC, pictorial blood loss assessment chart; PIH, Hypertensive disorders of pregnancy; PPH, postpartum hemorrhage; PPROM, preterm premature rupture of membranes; PR, pregnancy rate; PTB, Preterm birth; RDS, respiratory distress syndrome; SPTB, Spontaneous preterm birth; SVD, spontaneous vaginal delivery; VLBW, very low birth weight; VMS, vasomotor symptom.

**Table 3 jcm-12-00446-t003:** Summary of the study characteristics and the number and type of (un)rated outcomes of systematic reviews of nonrandomized studies (*n* = 18) that rated the outcome-specific certainty of evidence with GRADE.

Author, Year	Intervention/Exposure	Number of Studies	Number of Participants	Number of Outcomes Rated (Number of Unrated Outcomes)	Number of Rated Comparisons	Outcomes Rated
Mowat et al., 2016 [85]	The volume of surgeons in Gynecology surgery (low volume surgeons)	14	741,760	22	1	Total complications; Total complications adjusted OR; Total complications adjusted OR excluding gynecologic oncology; Intraoperative complications; Intraoperative complications adjusted OR; Intraoperative complications adjusted OR excluding gynecologic oncology; Postoperative complications; Postoperative complications adjusted OR; Postoperative adjusted OR excluding gynecologic oncology; Mortality; Medical complications; Operating time (mins); Transfusion; Estimated blood loss; Cystotomy; Ureteric Injury; Cystotomy or ureteric injury; Bowel injury; Vascular injury; Readmission; Reoperation; LOS > 2 days
Zafer et al., 2016 [86]	Semen washing followed by IUI, IVF, or IVF/ICSI.	40	4257 couples	1	1	HIV seroconversion
Martins et al., 2016 [87]	Cleavage stage; blastocyst transfer.	12	195,325	23	1	Perinatal mortality; Birth defects; PTB (<37 w); VPTB (<32 w); LBW (<2.5 kg); VLBW (<1.5 kg); BW > 4.0 kg; BW > 4.5 kg; GA; LGA; PE/PIH; GDM; PP; PAb; PAc; PROM; APH; PPH; CS; Low Apgar 5 min; Miscarriage; Stillbirth; VT
Park et al., 2016 [88]	Corticosteroid Therapy	17	3646	7	1	Mortality before discharge; RDS; IVH; NEC; Chronic lung disease; ROP stages greater than 2; Neurologic impairment 18–22 month
Wang et al., 2017 [89]	Fine needle aspiration cytology and core needle biopsy in evaluation of suspicious breast lesions	12	1802	2	1	True positives; False negatives; True negatives; False positives
Meireles et al., 2017 [90]	Metformin	19	unclear	3	1	Atypical endometrial Hyperplasia; Biomarkers proliferation; Overall Survival
Berg et al., 2018 [101]	3 (defibulation; antepartum defibulation; previous defibulation)	62	5829	6	3 (defibulation vs. no defibulation; antepartum defibulation vs. intrapartum defibulation; previous defibulation vs. antenatal defibulation)	CS; episiotomy; 2nd degree tear; 3rd degree tear; 4th degree tear; 1 min Apgar < 7
Cavoretto et al., 2018 [91]	IVF/ICSI	15	61,677	2	1	sPTB < 37 w; sPTB < 34
Alviggi et al., 2018 [92]	blastocyst-embryo transfer	14	197,275	8	1	PTB < 37 w; LBW (<2500 g); VPTB < 32 w; VLBW (<1500 g); SGA; LGA; perinatal mortality and congenital anomaly
Cai et al., 2019 [93]	3 (Rotating shiftwork, Fixed night shift, long working hours)	62	196,989	7	1	PTL; LBW; SGA; miscarriage; preeclampsia; gestational hypertension; IUGR
Haahr et al., 2019 [94]	Bacterial vaginosis	12	2980	6 (2)	1	LBR; early spontaneous abortion rate; CPR; Biochemical pregnancy rate; Implantation rate; PTL < 37 w
Saccone et al., 2019 [95]	Interventional radiology	15	958	4	1	Estimated Blood Loss; Units of Packed red blood cells transfused; Blood Loss > 2.5 L; Red Blood Cells Transfused > 5 Units
Vermey et al., 2019 [96]	ART	33	6,178,944	4	4 (ART vs. SC; non-ART (unspecified); non-ART (sub-fertile). FET-ART (frozen-embryo transfer) vs. SC)	Placenta preavia; placenta abruption; morbidly adherent placenta; abnormal cord insertion
Cai et al., 2020 [84]	Occupational activities	80	853,149	13	1	PTL (adjusted for confounders; unadjusted for confounders); LBW (adjusted for confounders; unadjusted for confounders); SGA (adjusted for confounders; unadjusted for confounders); Miscarriage (adjusted for confounders; unadjusted for confounders); Pre-eclampsia (adjusted for confounders; unadjusted for confounders); Gestational hypertension (adjusted for confounders; unadjusted for confounders); Intrauterine growth restriction
Backes et al., 2020 [97]	Resuscitation and intensive care	31	2226	3	1	Prevalence of survival; survival without major morbidity; prevalence of survival without moderate or severe neurologic impairment
Giorgione et al., 2020 [98]	Pregnancy complicated by preeclampsia	15	unclear	2	1	Incidence of hypertension after hypertensive disorders of pregnancies; incidence of hypertension after preeclampsia
Varghese et al., 2020 [99]	Neoadjuvant chemotherapy	17	3249	8	1	Overall complication; Flap losses (total and partial) in free-flap reconstruction; Flap losses (total and partial) for all autologous flaps (pedicled and free-flaps); Implant or Expander losses; Hematomas in all reconstructions; Seromas in all reconstructions; Wound complications in all reconstructions; Delay to adjuvant therapy
Joyeux et al., 2020 [100]	Open fetal surgery for spina bifida aperta	17	unclear	13	1	Maternal death; Postoperative death ≤ 7 d; Mean operation time (min); Technical failure; PPROM < 30 + 0 w; Delivery < 30 + 0 w; In utero complete reversal of HH; Any treatment at repair site; Additional recovery at repair site; Improved motor function at birth; Complete reversal of HH at 12 months; CSF diversion at 12 months; Improved motor function at 30 months

Exposure: ART, Assisted reproduction technology; ICSI, intracytoplasmic; IUI, intrauterine insemination; IVF, in vitro fertilization. Outcome graded: APH, antepartum hemorrhage; BW, birth weight; CS, cesarean section; CSF, cerebrospinal fluid; CPR, Clinical pregnancy rate; FET-ART, frozen-embryo transfer; GA, gestational age; GDM, gestational diabetes mellitus; HH, hindbrain herniation; HIV, human immunodeficiency virus; IUGR, intrauterine growth restriction; IVH, Intraventricular hemorrhage, LBR, Live birth rate, LBW, low birth weight; LGA, Large for gestational age; LOS, Length of stay; NEC, necrotizing enterocolitis; PAc, placenta accrete; PAb, placental abruption; PE, pre-eclampsia; PIH, Hypertensive disorders of pregnancy; PP, placenta previa; PPH, postpartum hemorrhage; PPROM, preterm premature rupture of membranes; PROM, preterm rupture of membranes; PTB, Preterm birth; PTL, Preterm delivery; PTL, Preterm delivery; RDS, respiratory distress syndrome; ROP, retinopathy of prematurity; SGA, Small for gestational age; SPTB, Spontaneous preterm birth; VLBW, very low birth weight; VPTB, very preterm birth; VT, vanishing twin.

**Table 4 jcm-12-00446-t004:** Summary of the study characteristics and the number and type of (un)rated outcomes of systematic reviews of RCTs/NRSs (*n* = 8) that rated the outcome-specific certainty of evidence with GRADE.

Author, Year	Intervention/Exposure	Study Design	Number of Studies	Number of Participants	Number of Outcomes Rated (Number of Unrated Outcomes)	Number of Rated Comparisons	Outcomes Rated
Armstrong et al., 2017 [103]	Baseline anatomical assessment of the uterus and ovaries in infertile women	RCTs/NRSs (separately)	19	unclear	3	1	Diagnostic value of a baseline assessment of the uterus and ovaries with imaging or surgery; versus pelvic examination; Diagnostic value of transvaginal ultrasound; sonohysterography; hysteroscopy; HSG; HyCoSy and MRI in detecting uterine; endometrial and/or ovarian pathology; Prognostic value of assessment of the uterus and ovaries with imaging or surgery; on clinical pregnancy and live birth (became pregnant unassisted or through ART);
Shan et al., 2020 [109]	Gabapentin and pregabalin	RCTs and NRSs (separately)	21	3519	24	3 (placebo; estrogen; antidepressants)	HF frequency (4 weeks; 8 weeks; 12 weeks; 24 weeks); HF duration (4 weeks; 12 weeks); HF severity (composite score): (4 weeks; 8 weeks; 12 weeks); Dizziness; Drop out of AE; Fatigue; Headache; Insomnia; Nausea; Somnolence; Weigh gain
Bellos et al., 2020 [104]	Non-steroidal anti-inflammatory drugs	RCTs and NRSs (combined)	10	1647	4	1	Severe hypertension; Systolic blood pressure; Diastolic blood pressure; Mean arterial pressure
Zhang et al., 2020 [108]	Sacrocolpopexy (transvaginalmesh surgery)	RCTs/NRSs (separately)	20	unclear	12 (3)	1	Anatomical success; Total vaginal length; Point C site; Subjective success; Mesh-related complications; Reoperation for prolapse recurrence; De novo dyspareunia; Operating time; Bladder injury; Bowel injury; Hematoma.
Grabovac et al., 2018 [106]	Caesarean section	RCTs/NRSs (NRS)	15	12,335	7	1	Death (23–27 + 6 w; 23- 24 + 6 w25; 26 + 6 weeks; 27–27 + 6 w); IVH gr III/IV (23–27 + 6 w; 25–26 + 6 w; 27–27 + 6 w)
Maheshwari et al., 2018 [107]	Frozen embryo	RCTs/NRSs (NRS)	26	940,047	5	1	SGA; BW < 2500 gm (low birth weight); LGA; PTL; PIH
Nassr et al., 2017 [104]	Vesico-amniotic shunt	RCTs/NRSs (combined)	9	246	4	2	Perinatal survival; 6–12 month postnatal survival; 2-year postnatal survival; renal function of fetuses
Kim et al., 2017 [105]	Self-administration of injectable contraceptives	RCTs/NRSs (separately)	3	264	10 (2)	1	Vontinuation (non-interrupted use at 12 months; non-interrupted use at 3 months); Satisfaction with the contraception method (want to continue this method) at 12 months; Satisfaction with the contraception method (prefer to continue the method) at 3 months; Satisfaction with the contraception method (recommend to a friend) at 12 months; Satisfaction with the contraception method at 3 months; Satisfaction in location (at 3 months)

Study design: RCTs, randomized controlled trials; NRSs, nonrandomized studies. Outcome graded: AE, Adverse Event; ART, assisted reproduction technology; BW, birth weight; HF, hot flash; HSG, hysterosalpingography; IVH, Intraventricular hemorrhage; LGA, Large for gestational age; PIH, Hypertensive disorders of pregnancy; PTL, Preterm delivery; SGA, Small for gestational age.

**Table 5 jcm-12-00446-t005:** Summary of the systematic review characteristics.

	Total SRs (*n* = 67)	SRs of RCTs (*n* = 41)	SRs of NRSs (*n* = 18)	SRs of RCTs/NRSs (*n* = 8)
No of primary studies, median (IQR)	15 (8–21)	10 (6–20)	17 (14–25)	17 (9–21)
No of participants, median (IQR)	3962 (1632–7238)	2976 (1533.5–5412.75)	5829 (2791–333,396)	2583 (260–244,263)
No of summary of findings table	60	39	14	7
No of meta-analysis conducted	64	41	17	6
No of outcomes rated in a SR, median (IQR)	5 (3–8)	4 (3–8)	6 (3–11)	6 (4–11)
Category	67	41	18	8
Drug therapy, *n* (%)	28 (41.79)	22 (53.66)	3 (16.67)	3 (37.50)
Surgical therapy, *n* (%)	10 (14.93)	6 (14.63)	2 (11.11)	2 (25.00)
Assisted reproductive, *n* (%)	9 (13.43)	3 (7.32)	5 (27.78)	1 (12.50)
Drug and Surgical therapy, *n* (%)	2 (2.99)	2 (4.88)	0	0
Screening method, *n* (%)	4 (5.97)	2 (4.88)	1 (5.56)	1 (12.50)
Special disease, *n* (%)	2 (2.99)	0	2 (11.11)	0
Reproductive strategy, *n* (%)	1 (1.49)	0	1 (5.56)	0
Lifestyle factors, *n* (%)	3 (4.48)	1 (2.44)	2 (11.11)	0
Clinical care, *n* (%)	1 (1.49)	0	1 (5.56)	0
Virus, *n* (%)	1 (1.49)	1 (2.44)	0	0
Others, *n* (%)	6 (8.96)	4 (9.76)	1 (5.56)	1 (12.50)

IQR: interquartile range; NRSs, nonrandomized studies; RCTs, randomized controlled trials; SRs, systematic reviews.

**Table 6 jcm-12-00446-t006:** Frequency of the rating domains according to study design.

	Total (%)	RCTs (%)	NRSs (%)	RCTs/NRSs (%)
The rating of the certainty per outcome, *n*	946	614	324	8 (2 study)
High, *n* (%)	110 (11.63)	103 (16.78)	7 (2.15)	0
Moderate, *n* (%)	167 (17.65)	147 (23.94)	19 (5.85)	1 (12.50)
Low, *n* (%)	269 (28.44)	201 (32.74)	66 (20.72)	2 (25.00)
Very low, *n* (%)	400 (42.28)	163 (26.54)	232 (71.58)	5 (62.50)
Total number of downgrading domains, *n*	1473	1027	427	19
Risk of bias, *n* (%)	492 (33.40)	334 (32.52)	150 (35.13)	8 (42.11)
Imprecision, *n* (%)	582 (39.51)	434 (42.26)	143 (33.49)	5 (26.31)
Inconsistency, *n* (%)	201 (13.65)	100 (9.73)	98 (22.95)	3 (15.79)
Indirectness, *n* (%)	122 (8.28)	104 (10.12)	15 (3.51)	3 (15.79)
Publication bias, *n* (%)	76 (5.16)	55 (5.36)	21 (4.92)	0
Total number of upgrading domains, *n*	42	0	42	0
Large effect, *n* (%)	25(59.52)	0	23 (54.76)	0
Dose-response, *n* (%)	1(2.38)	0	1 (2.38)	0
Plausible confounding, *n* (%)	18(42.86)	0	18 (42.86)	0
Unclear, *n* (%)	0	0	0	0
Frequency of the rating domains				
Mean frequency, *n* of Downgrading domains, *n*/ The rating of the certainty per outcome, *n*	1.56 1473/946	1.67 1027/614	1.32 427/324	2.38 19/8
Risk of bias, *n* (% of outcomes downgraded)	492 (52.01)	334 (54.40)	150 (46.30)	8 (100.00)
Imprecision, *n* (% of outcomes downgraded	582 (61.52)	434 (70.68)	143 (44.14)	5 (62.50)
Inconsistency, *n* (% of outcomes downgraded)	201 (21.25)	100 (16.29)	98 (30.25)	3 (37.50)
Indirectness, *n* (% of outcomes downgraded)	122 (12.90)	104 (16.94)	15 (4.63)	3 (37.50)
Publication bias, *n* (% of outcomes downgraded)	76 (8.03)	55 (8.96)	21 (6.48)	0
Mean frequency, *n* of Upgrading domains, *n*/ The rating of the certainty per outcome, *n*	0.05 42/946	0 0/614	0.13 42/324	0
Large effect, *n* (% of outcomes upgraded)	25 (2.64)	0	23 (7.10)	0
Dose-response, *n* (% of outcomes upgraded)	1 (0.11)	0	1 (0.31)	0
Plausible confounding, *n* (% of outcomes upgraded)	18 (1.90)	0	18 (5.56)	0
Unclear, *n* (% of outcomes upgraded)	0	0	0	0

NRSs, nonrandomized studies; RCTs, randomized controlled trials; SRs, systematic reviews.

### 3.4. Upgrading and Downgrading Domains

The reasons for downgrading and upgrading are shown in Table 6. In total, 1473 instances of downgrading were identified, namely due to imprecision (39.51%), risk of bias (RoB) (33.40%), inconsistency (13.65%), indirectness (8.28%), and publication bias (5.16%), as well as 44 instances of upgrading due to large effect (59.52%), plausible confounding (42.86%), and dose-response (2.38%). Downgrading for imprecision and indirectness was more common in the SRs of RCTs (42.26%, 10.13%) compared to the SRs of NRSs (33.49%, 3.51%), whereas downgrading for RoB and inconsistencies was more common in the SRs of NRSs (35.13%, 22.95%), compared to the SRs of RCTs (32.52%, 9.74%). In addition, upgrading for dose-response, large effect, and plausible confounding were all in the SRs of NRSs.

We counted an approximate mean of 1.67 downgrades per outcome in the SRs of RCTs and a mean of 1.32 downgrades per outcome in the SRs of NRSs. The downgrading frequency (the number of downgrades per the number of rated outcomes) in the SRs of RCTs was higher for imprecision (70.68%), RoB (54.40%), indirectness (16.94%), and publication bias (8.96%), and lower for inconsistency (16.29%), compared to the downgrading frequencies encountered in the SRs of NRSs. Besides, 13.00% of outcomes rated to the SRs of NRSs were upgraded, 7.10% outcomes were upgraded for large effect, 5.56% for plausible confounding, and 0.31% for dose-response. Of note, no outcome was upgraded for unclear effect (Table 6). The reasons for the authors’ choice to downgrade or upgrade the certainty of evidence for outcomes are summarized in Appendix A.

## 4. Discussion

### 4.1. Principal Findings

To the best of our knowledge, this is the first study to examine the extent to which GRADE has been used in SRs published in the top 10 gynecology and obstetrics journals with the highest impact factor, according to the JCR 2019. In the last five years, the number of SRs using GRADE to evaluate the certainty of evidence was relatively small, but the number of it showed an upward trend, reaching 18 (7.56%) in 2020. In general, this methodological survey shows there were only 67 SRs (7.04%) that rated the outcome specific certainty of evidence with GRADE. Four hundred (42.28%) and 269 (24.88%) individual outcomes were rated as very low and low, respectively. In the SRs of RCTs, the certainty of evidence was downgraded mostly for RoB and imprecision, while in the SRs of NRSs, the certainty of evidence was downgraded mostly for RoB, imprecision, and inconsistency.

It is a very important finding that such a low proportion of evidence evaluated using GRADE in gynecology and obstetrics is of high quality. There are several reasons for this. First, several limitations in clinical studies such as the lack of a clearly randomized allocation sequence, blinding, allocation concealment, and failure to adhere to intention-to-treat analysis are inevitable, and can lower the quality of the evidence. In our study, we found all these accounting for 33.40%, among downgrading domains. Second, in clinical studies, it can be difficult to control the number of participants. If the number of participants is low and the confidence interval is wide, the quality of the evidence will be downgraded due to imprecision. For example, in a RCT conducted by Armstrong, et al. [103], the number of women undergoing hysteroscopy after IVF (in vitro fertilization) failure was low, so the quality of evidence was downgraded for impression. In addition, indirectness, publication bias, and inconsistency are also reasons for such low quality.

### 4.2. Results in the Context of What Is Known

This is the first scoping review on the use of GRADE in gynecology and obstetrics, and other publications have also addressed the details, advantages, and challenges of the GRADE approach in the clinical field [110,111,112]. As noted in the findings of the scope review above, certainty of evidence in NRSs may be downgraded too much, which would appear to prevent the application of GRADE in such cases. However, solutions to this challenge may help to promote the use of the GRADE approach. For example, researchers have recommended using ROBINS-I in GRADE. ROBINS-I offers an alternative terminology: establishing NRS rather than observational studies, which will give researchers a more transparent way of separating studies by design. This placed RCTs and NRS on a common metric for risk of bias, and facilitated the comparison of evidence from both types of studies.

Considering that the research design is the initial and key point of GRADE, in determining the level of evidence, our research investigated the use of GRADE in the evaluation of evidence in gynecology and obstetrics, and additionally presented the use of down- and upgrading factors, according to the study design reviewed in the SRs. The results of our study were similar to findings reported by Cuello-Garcia and colleagues [113], which showed that most respondents would present pooled data from both RCTs and NRSs separately, either in a single Summary of Findings Table or each in its own table. As our sample size is small, these findings need to be supported by testing on a larger sample.

### 4.3. Implications for the Broader Research Field

Having been adopted by more than 100 organizations worldwide indicates that GRADE is a promising approach to evaluating the certainty of evidence and determining the strength of a recommendation. Ref. [11] However, our experience permits us to put forward a few recommendations on assessing certainty of evidence and strength of recommendations.

First, GRADE has been widely used for rating the level of evidence from both RCTs and NRS, but using GRADE to evaluate the certainty of evidence in NRSs continues to present some challenges. Since the study design is the initial and key consideration of GRADE to rate the initial certainty of evidence, RCTs start as “high”, whereas NRSs start as “low”, due to the risk of confounding and selection bias, users may improperly double count the risk of confounding and selection bias, so the certainty of evidence in NRSs may be downgraded excessively. Several opportunities for GRADE are presented by ROBINS-I (risk of bias in non-randomized studies of interventions) [114]. Since ROBINS-I places RCTs and NRS on a common metric for risk of bias, it may facilitate the comparison of evidence from both types of studies [114]. ROBINS-I does not consider study design as a risk of bias, such as cohort, case-control, case series, or cross sectional. The GRADE certainty of evidence from a body of studies using NRS designs would be high when ROBINS-I is used to assess risk of bias in NRS, since selection bias and confounding are assessed as integral components of ROBINS-I [114]. In addition, ROBINS-I provides a way to assess whether failure to use randomization in individual studies impacts bias risk and harmonizes GRADE approaches for different types of questions, such as prognosis and test accuracy [114]. In order to accurately rate the certainty of a body of evidence, it is highly recommended to use ROBINS-I in GRADE.

Second, while GRADE provides a framework for a systematic, transparent, and explicit assessment of the certainty of evidence and strength of recommendations, using GRADE will commonly involve some subjective judgments, and result in various assessments [115,116]. For example, if the GRADE users are uncertain about the exact ratings for multiple domains or when the same issue affects multiple domains, they may render various judgments. In this case, GRADE users should consider the domains together and choose the worst rating considered in one domain and the best rating considered in the other. Further, transparency requires presenting the reasoning for all judgments. Third, for better recommendations, individual patient conditions, preferences, and values should also be considered in addition to the certainty of evidence, as well as other important factors in GRADE approach.

## 5. Strengths and Limitations

Our study has several strengths. Firstly, we performed a rigorous screening and extensive data extraction, with information such as the number of outcomes rated and unrated outcomes, number and reason identified for down- and upgrading domains being included. Secondly, this is the first study to examine the extent to which GRADE has been used in SRs published in gynecology and obstetrics journals, which may contribute to improving the process of evidence-informed diagnostic methods in this area.

Limitations of this study are as follows. Firstly, our study focused on SRs published in the top 10 gynecology and obstetrics journals within a time frame limited to 5 years, which consequently resulted in a relatively small sample size (*n* = 67) and lack of inclusion in other medical and lower-ranked gynecology and obstetrics journals. Nevertheless, SRs published in high-impact medical journals are more likely to be relevant for future research and to provide evidence for clinical practice. Second, prior to this study, no official research protocol about GRADE use had been published, therefore, we reported our findings following the Preferred Reporting Items for Systematic Reviews and Meta-Analyses (PRISMA) framework where applicable to study design. Third, our study was based on a descriptive examination of GRADE use in SRs in gynecological and obstetric, rather than determining whether the authors followed criteria conducted by the GRADE working group to rate the certainty of the outcomes. Our study indicates that some authors may misconstrue GRADE domains. For example, the certainty of evidence in SRs was upgraded due to low RoB, narrow confidence intervals, very low P-values and mild statistical heterogeneity, rather than upgrading domains. Therefore, future research should give priority to the optimal use of the GRADE approach. Finally, no time trends are addressed or assessed in this report. GRADE system adoption has possibly increased and improved over time.

## 6. Conclusions

This methodological investigation attempts to reveal the application of GRADE in gynecology and obstetrics. As an explicit, comprehensive, transparent, and pragmatic evaluation system, GRADE gives the strength of evidence and recommended significance for each outcome, which makes it important to evaluate the certainty of the review evidence in obstetrics and gynecology. Our study shows that the use of rating the certainty of evidence in gynecology and obstetrics SR is relatively few. More attention should be paid to the use of ROBINS-I in GRADE, to the transparency of GRADE in the evaluation of evidence, and the actual situation of patients in the synthesis of gynecology and obstetrics evidence, to provide evidence for the final decision of clinical researchers and clinicians.

## Figures and Tables

**Table 1 jcm-12-00446-t001:** The distribution of systematic reviews (*n* = 67) that rated the outcome-specific certainty of evidence with GRADE by year and journal.

	Number of SRs Published	Number of SRs Rating the Outcome Specific Certainty of Evidence with GRADE, *n* (% of SRs Published)	Impact Factor
**Total, *n***	952	67 (7.04%)	-
2016	196	12 (6.12%)	-
2017	195	12 (6.15%)	-
2018	163	15 (9.20%)	-
2019	160	10 (6.25%)	-
2020	238	18 (7.56%)	-
Journal			
Ultrasound in Obstetrics & Gynecology	119	21 (17.65%)	8.678
British Journal of Obstetrics and Gynecology	244	17 (6.97%)	7.331
American journal of obstetrics and gynecology	117	8 (6.84%)	10.693
Human Reproduction Update	93	8 (8.60%)	17.179
Fertility and Sterility	90	5 (5.56%)	7.490
Breast	65	4 (6.15%)	4.254
Human Reproduction	45	2 (4.44%)	6.353
Obstetrics & Gynecology	88	1 (1.14%)	7.623
Gynecologic Oncology	82	1 (1.22%)	5.304
Best Practice & Research Clinical Obstetrics & Gynecology	8	0	4.268

SRs, systematic reviews.

## Data Availability

Data is contained within the article or Appendix A.

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
