# Peer review of "GRADE Use in Evidence Syntheses Published in High-Impact-Factor Gynecology and Obstetrics Journals: A Methodological Survey"

_jcm, 2023, doi:10.3390/jcm12020446_

Round 1
Reviewer 1 Report
Please include in the introduction previous methodological surveys for GRADE use in evidence syntheses. Please discuss how this technique was used in other fields of medicine.
Why did the author only use Pubmed? There are other databases with larger sets of research publications.
What is the reason for including systematic reviews published from 2016 to 2020?
Why did the authors only include the top 10 journals in obstetrics and gynecology?
Why did the authors use JCR 2019 and not the latest JCR impact factor report?
For the results, the authors can classify the SRs whether they are under obstetrics or under gynecology. It would be interesting to see if there are discrepancies and differences between these two subfields in OBGyn.
For Table 1, please remove the column “journal category”. That should be a given because this manuscript included obstetrics and gynecology journals.
For Table 5, please indicate what the values within and outside the parentheses mean.
Please correct minor grammatical errors within the manuscript.
The authors interchangeably used non-randomized and nonrandomized. Please choose one and use it in the manuscript.
Lines 35 to 37 Please cite references.
Lines 146 to 155 Please check the given IQR. Since this is a range, there should be two numbers indicated here.
Line 270 Please change “scope review” to “scoping review”.
Lines 272 to 274 Please rewrite this sentence because it is confusing.
Line 303 Please change “providing” to “provides”.
Please fix the section on strengths and limitations. For example, this was included as strength and limitation for the manuscript: “all published SRs using GRADE in the top 10 gynecology and obstetrics journals 316 published between 2016 and 2020”.
Reviewer 2 Report
Article is about rate of certainty evaluation of OG articles using GRADE approach. The novelty of the subject is welcomed and it is well structured.
I recommend rephrasing the passage "More attention needs to be paid to strengthening GRADE" with a more clear take home message for researchers.
Author Response
Response to Reviewer 2 Comments
Point 1: I recommend rephrasing the passage "More attention needs to be paid to strengthening GRADE" with a more clear take home message for researchers.
Response 1: Thanks for your kindly comments. We have rewritten this sentence as following:
“More attention should be paid to the use of ROBINS-I in GRADE, to the transparency of GRADE in the evaluation of evidence and the actual situation of patients in the synthesis of gynecology and obstetrics evidence to provide evidence for the final decision of clinical researchers and clinicians.” (See: Lines 355-359)

Reviewer 3 Report
Thank you for the opportunity to review this study.
The review is focused on GRADE use in evidence syntheses published in high-impact factor Gynecology and Obstetrics journals.
In general, your study is very interesting
In the sentence line 35, the citations are not in correct format.
In line 72, where is referred that the study includs the top 10 published gynecology and obstetrics journals between 1 January 2016 to 31 December 2020, with the highest impact factor according to the JCR 2019, authors could write the range of impact factors of the journals.
Also, in line 272 the authors refer to solutions on how the GRADE approach can be applied in clinical field and promote the use of this, we suggest to the authors to analyze briefly these solutions.
Last, in line 296 is mentioned ROBINS-Ι, authors could analyze briefly how ROBINS-Ι works and how accurately rate the certainty of a body of evidence.
Generally, authors can briefly and clearly, in the end explain why is important the use of GRADE approach to identify and describe the certainty of evidence of gynecology and obstetrics systematic reviews.
Author Response
Response to Reviewer 3 Comments
Point 1: In the sentence line 35, the citations are not in correct format.
Response 1: Thanks for your kindly comments. We have changed the citations in correct format.
“Systematic reviews (SRs) are essential parts of evidence-based medicine and serve as the basis for clinical practice guidelines,1 also widely used in the field of gynecology and obstetrics.2-6” (See: Lines 37-39)
Point 2: In line 72, where is referred that the study includs the top 10 published gynecology and obstetrics journals between 1 January 2016 to 31 December 2020, with the highest impact factor according to the JCR 2019, authors could write the range of impact factors of the journals.
Response 2: Thanks for your kindly comments. We have added the range of impact factors of the journals as following:
“Systematic reviews (SR) published between 1 January 2016 to 31 December 2020 in the 10 gynecology and obstetrics journals with the highest impact factor (range: 17.18-4.25) according to the JCR 2019 were identified through searches in the database PubMed. (Appendix 1).” (See: Lines 78-81)
Point 3: Also, in line 272 the authors refer to solutions on how the GRADE approach can be applied in clinical field and promote the use of this, we suggest to the authors to analyze briefly these solutions.
Response 3: Thanks for your kindly comments. We have analyzed briefly these solutions as following:
“However, solutions to this challenge may help to promote the use of GRADE approach. For example, researchers have recommended using ROBINS-I in GRADE. ROBINS-I offers an alternative terminology: establishing NRS rather than observational studies, which will give researchers a more transparent way of separating studies by design. This placed RCTs and NRS on a common metric for risk of bias and facilitated the comparison of evidence from both types of studies.” (See: Lines 274-280)
Point 4: Last, in line 296 is mentioned ROBINS-Ι, authors could analyze briefly how ROBINS-Ι works and how accurately rate the certainty of a body of evidence.
Response 4: Thanks for your kindly comments. We have added how ROBINS-Ι works and how accurately rate the certainty of a body of evidence as following:
“ROBINS-I does not consider study design as a risk of bias, such as cohort, case-control, case series, or cross sectional. The GRADE certainty of evidence from a body of studies using NRS designs would be high when ROBINS-I is used to assess risk of bias in NRS, since selection bias and confounding are assessed as integral components of ROBINS-I.” (See: Lines 304-308)
Point 5: Generally, authors can briefly and clearly, in the end explain why is important the use of GRADE approach to identify and describe the certainty of evidence of gynecology and obstetrics systematic reviews.
Response 5: Thanks for your kindly comments. We have added the information in the revised manuscript as following:
“As an explicit, comprehensive, transparent, and pragmatic evaluation system, GRADE gives the strength of evidence and recommended significance for each outcome, which makes it important to evaluate the certainty of the review evidence in obstetrics and gynecology.” (See: Lines 351-354)

Reviewer 4 Report
The manuscript is clear and scientifically sound. The topic is relevant for the field of evidence-based medicine. The results are presented and interpreted appropriately. The conclusions are coherent.
To help potential readers to assess the significance of the presented GRADE system for gynecology and obstetrics, the author may add an overview on its use in other fields of clinical medicine as well as in non-medical areas of science.
Round 2
Reviewer 1 Report
The authors have addressed all my comments. The manuscript is now acceptable for publication in its current form.